# An Overview of the Circadian Clock in the Frame of Chronotherapy: From Bench to Bedside

**DOI:** 10.3390/pharmaceutics14071424

**Published:** 2022-07-06

**Authors:** Alan Vandenberghe, Marc Lefranc, Alessandro Furlan

**Affiliations:** 1Univ. Lille, CNRS, UMR 8523-PhLAM-Physique des Lasers Atomes et Molécules, F-59000 Lille, France; alan.vandenberghe@univ-lille.fr; 2Tumorigenesis and Resistance to Treatment Unit, Centre Oscar Lambret, F-59000 Lille, France; 3Univ. Lille, CNRS, Inserm, CHU Lille, UMR9020-U1277-CANTHER-Cancer Heterogeneity Plasticity and Resistance to Therapies, F-59000 Lille, France

**Keywords:** circadian clock, pathologies, metabolic disorders, mathematic modeling, chronotherapy

## Abstract

Most living organisms in both the plant and animal kingdoms have evolved processes to stay in tune with the alternation of day and night, and to optimize their physiology as a function of light supply. In mammals, a circadian clock relying on feedback loops between key transcription factors will thus control the temporally regulated pattern of expression of most genes. Modern ways of life have highly altered the synchronization of human activities with their circadian clocks. This review discusses the links between an altered circadian clock and the rise of pathologies. We then sum up the proofs of concept advocating for the integration of circadian clock considerations in chronotherapy for health care, medicine, and pharmacotherapy. Finally, we discuss the current challenges that circadian biology must face and the tools to address them.

## 1. Introduction

The circadian clock orchestrates most physiological events in living organisms and its deregulation in association with modern ways of life correlates with the rise of multiple pathologies in humans. Interestingly, growing evidence opens perspectives to improve human health via intervention on the circadian clock circuitry.

Many excellent reviews are available on the several facets of circadian rhythms in distinct physiological processes (sleep, metabolism, physiology, immunity, …) [1,2,3,4,5,6,7]. Still, most of them focus on peculiar aspects of a specific field. In this review, we try to outline integrated data dealing with experiments, animal models, mathematical models, and clinical applications to give an overview of the multiple assets circadian medicine possesses to improve human health.

## 2. A Brief History of Circadian Rhythm Knowledge

Circadian rhythms in nature were observed for a long time, but their importance in physiology has come back into the spotlight recently. The circadian clock has gained much notoriety with the Nobel Prize in Physiology or Medicine awarded to Jeffrey Hall, Michael Rosbash, and Michael Young for their studies on the molecular mechanisms controlling the circadian rhythms [8,9,10].

In the plant kingdom, Androsthenes made the first report on circadian rhythmicity in the 4th century BC with the sleep movements of Tamaricus indicus’ leaves. Some two millennia occurred before some experimental work reinforced this concept of the plant circadian clock. In 1729, Jean-Jacques d’Ortous de Mairan actually noticed that the Mimosa leaves moved with a 24 h periodicity, even when he put the plant in a light-deprived environment [11]. This proved that an endogenous clock was able to impose a rhythm to processes, which could be discriminated from simple responses to daily stimuli.

Circadian rhythmicity is also obviously present in the behavioral patterns of animals (sleeping and feeding notably) and leads to the classification between diurnal and nocturnal animals. Besides, many biological processes including body temperature, hormone release, immune response, …, are now recognized to follow circadian evolutions.

Since almost all living organisms display circadian rhythms, this probably implies that evolution has selected the preservation of such rhythms. By allowing organisms to anticipate and prepare for circadian environmental changes, they would provide a selective advantage to living beings to optimize the light and food resources.

The molecular mechanisms responsible for these circadian rhythms remained elusive until the development of mutation screenings in the genome of the fruit fly a few decades ago. At that time, a seminal work highlighted the *period* (*per*) gene as critical for the periodicity of the fly activity [12]. Since then, researchers identified several other genes as core clock genes, driving circadian rhythms. This clock is based on an intricate network of intertwined transcriptional feedback loops acting as endogenous oscillators [13] (Figure 1).

The molecular core clock relies on the inter-regulation of several genes, including the couple Bmal1/Clock and their transcriptional targets Per, Cry, Ror, and Rev-Erb. When Bmal1 and Clock proteins dimerize, they bind to E-box-containing promoters and activate the transcription of their targets. Among these, PER and CRY proteins dimerize and repress the binding of BMAL1/CLOCK to E-boxes. The expression of REV-ERB will repress the transcription of the *bmal1* gene, whereas ROR will stimulate this transcription, via their action on RRE (ROR Response Element) in the *bmal1* gene promoter. These intertwined negative and positive feedback loops ensure proper circadian oscillations in the levels of these gene products, with a peak in BMAL1/CLOCK activity at the onset of the active phase, and a more or less delayed expression of its targets/regulators.

Nowadays, researchers evaluate that more than 40% of protein-coding genes display 24 h rhythmicity in mammals [14]. Moreover, it seems that beyond transcription, circadian rhythms of oxidation and reduction occur, as evidenced in red blood cells [15]. A link between these two kinds of regulation may be the Pentose Phosphate Pathway, with NADPH production regulating core clock transcriptional circuits [16].

How to translate the great amount of knowledge emerging in the field of the molecular circadian clock to improve human health will be the topic of the following chapters in this review.

## 3. Alterations in Day–Night Lifestyle Influence on Human Health: Of the Misdeeds of Human Circadian Misalignment

The central circadian clock in humans is localized in the SCN (suprachiasmatic nucleus), which receives light-driven inputs from specific retinal photoreceptors. This central clock then coordinates the whole-body activities, notably by releasing hormones such as melatonin and glucocorticoids and via neurotransmitters, thus entraining peripheral clocks in the various tissues of the body [17] (Figure 2). The SCN thus regulates food intake, and can as well impact metabolism independently of feeding [18,19]. In parallel, peripheral tissues are also entrained by food intake, and restricted feeding can uncouple peripheral oscillators from the SCN pacemaker [20]. In rare conditions, peripheral clocks can even modulate the central clock to a certain extent [21].

The circadian central and peripheral clocks are entrained by two distinct main cues: light and food intake, respectively. The suprachiasmatic nucleus (SCN) is sensitive to light and synchronizes the central clock upon light exposure. This central clock transmits information to peripheral organs via the neuronal and endocrine systems and will regulate food intake and metabolism. In parallel, these peripheral organs also rely on food intake to entrain their clocks.

Proper synchronization of these biological oscillators, especially ensured by the absence of food intake in the normal rest phase of the organism, allows for optimized energy expenditure in most cellular and tissular processes. Inversely, clock misalignments generate troubles in metabolic functions, which can result in several pathologies.

For millennia, human beings have kept their activities rather aligned with cues from their environments (so-called zeitgebers) and their circadian clock. The invention of electricity and modern ways of life have shifted this principle in the last decades. Watching TV or computer screens at night has become increasingly common, and disturbs the integration of light cycles by our brain. At the same time, food intake tends to be less and less restricted to our active phase. Moreover, coffee consumption in the evening is becoming quite common and was shown to delay melatonin production and circadian rhythm, beyond its sleep disruptive effects [22,23].

Since circadian rhythmicity orchestrates most processes, misaligned rhythms are detrimental to one’s fitness. The circadian misalignment is especially exacerbated for some categories of people, either for deliberate leisure purposes or for imposed work reasons. Shifted workers thus represent a category of people regularly submitted to strong circadian misalignments and the impact of such practices on human health is illustrated by their increased propensity to develop several pathologies such as depression, metabolic syndrome, cardiovascular diseases, and cancers (for review, see [24]).

A clear experimental demonstration of a link between recurrent circadian clock shifts and pathologies came from a study in which mice were subjected to chronic jet lags. This led to a desynchronization of their SCN central clock, chronic inflammation, the development of hepatosteatosis, and a reduced lifespan [25].

Individual variations occur within the human population about the circadian clock. Some people naturally wake up early in the morning and are called “larks” or “early birds” whereas others are dubbed “night owls”. To quantify such phenotypes, researchers used the MSF (Midpoint Sleep on Free days) and showed that the chronotype evolved with age. Indeed, this MSF progressively delays reaching its maximal lateness around 20 years old, before decreasing regularly when becoming older [26]. Although this was considered mainly a question of education and choice in the way of life, recent studies suggest that this phenomenon may rely on genetic circuits, which are slightly shifting the body clock [27].

This state of things has consequences on people’s alertness, efficiency, and mood [28]. A very recent study has investigated the consequences of the shift between the late chronotype of most teenagers and the fact that they are attending school early in the morning. Of note, they show that school performance is improved when learning time is adjusted to the student/t’s chronotype [29]. This means that the performance assessment is somehow biased by one’s chronotype and possibly contributes to the fact that in our societies, the early bird catches the worm.

People who display a strong shift in their chronotype suffer from a so-called “social jetlag” [30]. Strikingly, this social jetlag is associated with depressive symptoms but also with cardiovascular dysfunctions [31]. Notably, later chronotypes correlate with increased psychological disorders, diabetes, poorer cardiovascular health [32], and an enhanced risk of mortality [33]. We will develop later on this association between the circadian clock and cardiovascular functions and diseases in the Section 6.

Another important circadian misalignment occurring in our modern societies is associated with the aging population. Older people tend to display dampened oscillations in several circadian traits, including the release of melatonin and cortisol hormones, which could be linked to some age-related pathologies (for review, see [34,35]).

## 4. The Molecular Links between the Circadian Clock and Metabolism: The Food Connection

The energetic demands of our tissues drastically vary night and day, and metabolic activities are accordingly fluctuating in a circadian manner. It is therefore not surprising that food intake represents a critical zeitgeber for peripheral clocks involved in metabolism regulation.

A clear proof of evidence of the link between circadian clock and metabolism came from studies showing that genetically disrupting either *Clock* or *Bmal1* in mice induced the apparition of several traits associated with metabolic syndrome [36,37].

Some nuclear receptors such as REV-ERB, ROR, and peroxisome proliferation-activated receptors (PPARs) both regulate cellular metabolism and are components of the molecular core clock machinery. This presence at the crossroad of metabolism and circadian clock makes them good candidates to control metabolic storage and energetic consumption accordingly to circadian activities [38]. Hence, REV-ERB agonists, such as SR9009, represent interesting pharmacological compounds due to their pleiotropic activities in the regulation of the circadian clock and lipid metabolism [39]. The key role of REV-ERB in the integration of clock and food inputs was recently reinforced by the demonstration that REV-ERBα cyclically modifies the activity of OGT (O-GlcNAc transferase), a nutrient sensor, which regulates among others the expression of SREBP1c, a critical protein in lipogenesis [40]. REV-ERBα also controls mTOR [41], another protein well known for integrating nutrient and energy status in the cells and recently shown to affect the circadian clock [42]. In addition, REV-ERBα regulates inflammation processes, and we will come back to this aspect in the chapter dealing with chronopharmacotherapy of inflammatory diseases.

To come back to the connection between metabolism and the clock, metabolite levels can as well modulate the clock and be controlled by it, as illustrated by the NAD^+^ (Nicotinamide Adenine Dinucleotide) example. NAD^+^ is a cofactor for SIRT1 deacetylase, which inhibits BMAL1 activity, and BMAL1/CLOCK conversely controls NAD^+^ level oscillations by controlling the NAD^+^ salvage pathway via NAMPT (nicotinamide phosphoribosyltransferase) expression control [43,44]. Moreover, SIRT1 deacetylates not only BMAL1 but also PGC1α, a transcriptional coactivator regulating energy homeostasis and the expression of core clock genes [45]. Since PGC1α activation depends on AMPK action before its subsequent deacetylation by SIRT1 [46], PGC1α stands as an integrator of both NAD^+^ and AMP energy-sensing inputs to control both circadian clock and metabolism.

It is noteworthy that circadian fluctuations in metabolites are coordinated between the different tissues of an organism, reflecting a coherence among clocks [47]. Furthermore, meal timing was shown to play a role in peripheral clock synchronization in humans [48]. Given this tight link between metabolism and the circadian clock, it is not surprising to notice that clock alterations result in metabolic pathologies. In the Gremlins movie, Gizmo’s seller tells the hero not to give him food after midnight. Human beings should also pay more attention to the recommendation not to eat during their resting phase. Actually, it seems that many disorders occurring following the mishandled circadian clock may rely on improper food intake timing, resulting in physiopathological situations. Experimental studies with mice demonstrated that shifting their feeding time to their circadian rest phase induced metabolic syndrome in these mice [49]. Changing one’s diet can reciprocally affect the circadian clock [50]. Indeed, mice that are given a high-fat diet shift their locomotor activities [51]. Moreover, the nature of food can also alter the circadian clock, as illustrated by BMAL1 inhibition triggered by the saturated fatty acid palmitate [52].

When modern ways of life disrespect both food intake and the circadian clock, several metabolic pathologies arise.

## 5. A Healthy Way of Life: Get Synchronized with Your Circadian Clock

After having shown that the timing of feeding can trigger great metabolic repercussions, mice models also demonstrated that good feeding habits could restore a healthy situation. Time-restricted feeding actually proved to prevent the onset of metabolic alterations in mice given a high-fat diet, for an equivalent caloric intake [53]. This method was efficient as well to counteract the metabolic diseases of clock-deficient mice [54].

With this in mind, and although evidence is still scarce for food as a *bona fide* zeitgeber in humans [55], several strategies of chronological diet are nowadays proposed to improve human metabolism and health. These include fasting periods of 12H or more, caloric restriction, or fasting-mimicking diets based on high fat and low carbohydrate contents [56,57]. Such strategies display the advantage to improve weight maintenance without requiring pharmacological drugs, yet the importance of each intervention remains debated since they are often combined in the dietary protocols. The use of natural bioactive compounds as diet supplements also drove promising results [58]. Notably, Epigallocatechin-3-gallate (EGCG) can restore a proper clock and metabolism in mice fed with a high-fat and high fructose diet, when administered at specific times [59].

Circadian clock misalignments are highly associated with negative health impacts on shift workers, who represent a large population in our modern societies. Rotating night shift work was especially shown to increase the risk of developing type 2 diabetes when it was combined with several unhealthy lifestyles [60]. Several research groups are trying to identify strategies aimed at alleviating the health problems met by this category of workers. A study performed with a small cohort of 11 individuals simulating the night shift suggested that the absence of eating at night may be helpful to impair the metabolic disruption in shift workers [61]. Although measuring metabolic traits requires complicated logistics in such studies, sleep improvements represent useful hints to evidence reduced misalignments in shifted workers. A chronotype-adjusted shift schedule was implemented in a factory, by removing the highest shifts for extreme chronotypes, which led to an increase in sleep duration and quality [62]. While it may be difficult to apply in most factories or hospitals, another approach could be implemented more easily on a large scale, which relies on light-based interventions and was also successful in improving sleep, as well as reducing fatigue and working errors [63].

Exposure to artificial light at night reduces melatonin secretion and alters sleep and circadian rhythms [64]. An easy tip for most people would therefore be to reduce this exposure as much as possible, by limiting the use of computer screens late at night. When not possible due to work duties, softwares exist which modify the spectrum of screen light day and night, and especially decrease the emission of blue light at night, with the intent to preserve the circadian clock. The pharmacological use of melatonin to counteract its reduced secretion could represent another solution [65], and we will come back to this point later on in this review.

More globally, the impact of the visual and non-visual effects of light on human activity is gaining importance nowadays, giving rise to the human-centric lighting concept [66] and recommendations such as adapting light environments throughout the day [67].

## 6. Circadian Medicine and Chronopharmacotherapy

Sleep alterations are common in hospital stays and are classically ascribed to a combination of a change in environment, noise, and disease. Sleep disturbances are associated with many deleterious effects on health [68], still, they are seen as normal in hospital practices and almost no effort is taken to reduce them as much as possible in the frame of inevitable nocturnal hospital routine. Interestingly, a study quantified low illuminance levels in patients’ rooms during the day, associated with half-closed rolling shutters, and showed that patients which were exposed to more light in the morning due to their location near the window slept better [69]. This suggests that simple attention paid to light delivery to the patients (window design/raising rolling shutters) could putatively improve their recovery. About the importance of sleep in disease management, it is also noteworthy that alterations in sleep and pain are connected, via the circadian clock and the release of melatonin and cortisol among others [70]. Finally, another intriguing question in sleep-associated pathophysiology resides in the misalignment of body clocks driven by obstructive sleep apnea, possibly via different tissue-specific answers to hypoxia [71].

The importance of circadian rhythms in cellular and molecular physiology is also gaining attention in the field of drug delivery [72] and has started to be seriously taken into account for translational research over the last years. Indeed, more and more studies call for the rise of a new facet of medicine called circadian medicine [73]. Circadian considerations significantly burden experimental and clinical settings, and this is in part why they had been neglected in our frantic race for new treatments. Yet they probably underlie the efficiency of many therapy protocols, depending on the rhythmicity of the process targeted.

About cardiovascular functions notably, a high amount of evidence is accumulating about circadian fluctuations in blood pressure, heart rate, platelet aggregation, and vascular endothelial function; and alterations in their rhythms can trigger the onset or progression of cardiovascular diseases [74]. It is now long known that acute myocardial infarctions occur more frequently in the morning [75]. This correlates with a peak in platelet activation during the morning, associated with the expression of REV-ERBα in platelets [76]. Anti-aggregative drugs such as acetylsalicylic acid (ASA) are therefore used for secondary prevention of coronary artery disease. Interestingly, a recent study showed that ASA administration in the evening improved platelet aggregation inhibition when compared to a morning administration [77]. In parallel, another study highlighted that a lack of antihypertensive drug administration specifically at bedtime was associated with increased mortality of patients [78], clearly claiming for chronotherapeutic considerations in cardiovascular disease care [79].

The tolerance of the myocardial tissue to ischemia/reperfusion highly varies within the day, as demonstrated in rodents [80], and this notion strikingly translated to human surgery a couple of years ago. The timing of heart surgery in humans was indeed shown to impact the risk of perioperative myocardial injury, with afternoon surgery leading to reduced troponin T release when compared to morning surgery [81]. This suggests that we have been missing critical physiological cues by overlooking chronobiological aspects, and raises meaningful questions about the organization of hospital interventions [82].

## 7. Clock, Immunity, and Inflammation

The circadian clock also applies to routine medicine practices, as illustrated by vaccination. When comparing the antibody titers following vaccinations against influenza realized either in the morning or in the afternoon, researchers highlighted that people who received the vaccine in the morning elicited a better antibody response (50). More globally, immunity follows circadian rhythms that will affect responses to pathogens, and circadian clock disruption can in addition influence the progression of autoimmune diseases [4]. Key molecular components of the clock take action in the immune response. REV-ERBα controls, for example, the production of proinflammatory cytokines after the exposure of pulmonary cells to LPS, a component of Gram-negative bacteria membrane [83]. Conversely, inflammatory factors can modulate the circadian clock, as illustrated by phase delays in clock oscillations induced by TNF-α and IL-1β cytokines [84]. Interestingly, and in link with the tight association between metabolism and clock, the palmitate fatty acid has a proinflammatory role, and its administration triggers phase shifting [85].

For viral infections, transgenic mice models deficient for BMAL1 displayed greater viral replications after infection with murid herpes or influenza viruses [86]. The authors of this study postulated that the seasonality in BMAL1 levels may contribute to the fact that infections with viruses such as influenza are more common in winter. Finally, SARS-CoV2 was shown to induce a change in the NAD gene set expression [87]. By using the murine hepatitis virus, another coronavirus model, the researchers showed that this kind of virus can deplete the cellular NAD^+^ stocks. This may counteract the innate response to these viruses by impairing the action of NAD^+^ dependent PARP enzymes, and the authors suggest that increasing NAD^+^ levels thanks to a NAMPT activator could be useful to restore the antiviral answer. In the context of COVID-19, low NAD^+^ concentrations were also associated with poor outcomes, and restoring higher levels is currently envisioned as a clinical strategy [88].

Asthma represents another respiratory pathology in which inflammation plays a key role. Asthma symptoms classically display a circadian rhythm, with a worsening of symptoms at night and in the early morning, possibly regulated by glucocorticoids, offering the possibility for chronopharmacology approaches [89]. More globally, alterations of the circadian clock have been highlighted in several chronic pulmonary diseases, including chronic obstructive pulmonary disease, cystic fibrosis, and pulmonary arterial hypertension, with particular involvement of REV-ERBα [90]. The control of inflammation by REV-ERBα was also evidenced in a mouse model of DSS-induced colitis, in which the administration of REV-ERBα agonist SR9009 was able to reduce the colic inflammation [91].

## 8. Clock and Age-Related Diseases: Molecular Studies Open Therapeutical Perspectives

The human lifespan has increased in the last decades, which raises major health issues related to many age-associated pathologies. As mentioned earlier in this review, aging is usually associated with disturbed circadian clocks [34]. Altered chronotypes of old people are also important to consider when assessing their cognitive functions, since their results may be much better in the early morning than in the late day [92]. Interestingly, it seems that healthy old people preserve a melatonin profile similar to young people [93]. This would mean that a decrease in the melatonin rhythm should not be regarded as a characteristic of aging but rather as a pathological feature. Accordingly, the pharmacological use of melatonin is now proposed to treat age-related pathologies, including Alzheimer’s disease (AD) [65]. The concept that a circadian clock disruption is occurring selectively in cortical and limbic circuits in AD patients is emerging, and strategies aimed at alleviating the pathogenesis through the restoration of a functional clock in these regions are under development [94]. Based on this concept, a study has, e.g., evidenced the therapeutic potential of a pharmacological approach using the clock modulator nobiletin [95]. Indeed, the administration of this natural compound to AD model mice dampened inflammation, astrogliosis, and β-amyloid deposition. Of note, disassociations between body temperature and locomotor activity rhythms were observed in the context of AD [96]. Such clock misalignments between different physiological rhythms were also reported in other pathologies, such as glaucoma where intraocular pressure rhythms were shown to be delayed when compared to those of body temperature as the disease progresses [97]. In this context, clock-based strategies look promising, as illustrated by the melatonin-mediated improvement of circadian rhythms and intraocular pressure in glaucoma patients [98].

Parkinson’s disease (PD) represents another age-associated pathology in which the circadian clock was recently evidenced to play a role, particularly in the mood disorders called “sundowning syndrome”. Researchers have reproduced such symptoms in a PD mouse model and have demonstrated that the delivery of a REV-ERBα antagonist, SR8278, could alleviate clock-associated depression and anxiety in these mice [99]. These examples thus highlight the potential of clock-related pharmacological approaches in the field of AD and PD.

Beyond neurodegenerative diseases, other approaches based on circadian regulation via the control of feeding also seem promising to improve a healthy lifespan [100,101]. At the molecular level, mice models revealed that a decrease in SIRT1 levels in the brains of aged mice drives clock alterations and may be important in aging [102]. In this context, SIRT1 and its partners represent promising molecular targets to restore physiological oscillations in old patients suffering from altered clock and/or metabolic dysfunctions. Since SIRT1 activity requires NAD^+^ as a cofactor, dietary supplementation with Nicotinamide riboside (NR) was proposed, yet much caution should be brought to the potential side effects of chronic use at high doses [103].

Altogether, these works pave the way for the integration of circadian clock considerations in chronotherapy for health care and medicine (Figure 3).

Many processes can disturb the circadian clock in humans. Aging is associated with an altered circadian clock for most people, but healthy old people seem to preserve a better circadian clock, which suggests that age-related health issues may rely on circadian clock alterations. Sleep alterations are common in modern societies, linked either with leisure or work. They can be associated with increased exposure to artificial light at night. Additionally, the poor quality of food composition and its improper timing disturb the clock. These three last events are particularly exacerbated in shift workers, who suffer frequent and abrupt changes in their clocks.

Bad physiological consequences of such processes can be counteracted by various clock-centered approaches. A properly-timed and balanced diet can improve clock oscillations and health, as well as the execution of physical exercise. Reducing exposure to artificial light at night improves sleep quality and melatonin rhythms. Medical practice efficacy also depends on the clock (vaccination elicits a better antibody response when realized in the morning, whereas performing heart surgery in the afternoon alleviates perioperative injuries). Finally, pharmacological drugs also present different efficacies depending on their administration time.

## 9. Current Challenges and Perspectives: How to Define the Most Appropriate Timing for Pharmacotherapy

Investigating the circadian clock imposes several constraints. The most obvious experimental aspect consists of the need for several measurements per day. To catch circadian fluctuations, a minimal number of six time points per day is usually required, and an increased frequency of time points will yield more accuracy in the determination of periods and amplitudes of the processes. When carried out with animal models, such experimental features quickly rise big issues in terms of ethics (if the measurement requires the euthanasia of the animals, one easily reaches several dozens of animals for a few conditions), costs, and in planning the team agenda. In that way, performing experiments with cell cultures allows reaching a much greater amount of information while minimizing these issues. Although cell cultures lack systemic inputs occurring in animals, they are based on the same molecular framework and provide measurements at the level of the single cell, which is unattainable in animals. For example, primary hepatocytes equipped with luciferase reporters allowed to investigate issues of clock synchronization between cells [104]. However, differences remain in the biological context between in vivo and in vitro experiments, such as the presence or absence of cofactors. This has generally cast doubt on the physiological relevance of cell culture experiments, even when placing cells in carefully controlled environments (3D cultures, organoids, …).

Another challenge to face when studying the molecular circadian clock is its great complexity. The core clock gene network consists of intertwined feedback loops with contradictory actions, and it is highly demanding to predict the outcomes of a perturbation on the whole circuitry. In this frame, a quantitative approach is needed to integrate a large number of experimental observations from the literature. A few years ago, we generated a mathematical model for the liver circadian clock with a peculiar interest in its modulation by food intake and metabolite flux [105]. This model pinpointed SIRT1 as a key element connecting the cell energy status NAD^+^ fluctuations during the day with PER-CRY, BMAL1-CLOCK, and PGC1α, and more importantly identified AMPK as a key metabolic driver of the clock. It had been previously reported that AMPK input to the clock proceeds through the control of CRY1 degradation [106], but the study by Woller et al. [105] suggested that the integration of AMPK and NAD^+^ signaling by PGC1α and the subsequent activation of BMAL1 play a more important role, highlighting PGC1α as a genuine player of the clock [107]. In particular, the model of Woller et al. could reproduce the decrease in clock amplitude observed upon a high-fat diet by simply decreasing AMPK activity. An intriguing finding of the same study was that delivering a REV-ERBα agonist at the right time could almost completely counteract the effect of AMPK depletion for most clock genes, suggesting a connection between these two key actors.

A more recent manuscript combined modeling and cellular experiments to get insight into the impact of the various activities of SIRT1 deacetylase on the circadian clock [108]. It nicely evidenced the prevalent importance of PER2 and PGC1α deacetylation by SIRT1 over that of BMAL1 in entraining the transcriptional clock, confirming some of the predictions of Woller et al. [105]. Interestingly, in the physiopathological context of aging that was previously discussed in this manuscript, simulations with mathematical models could recapitulate the circadian clock alterations associated with aging by integrating the reduced SIRT1 levels that were observed in aged individuals [109].

Altogether, these studies highlight how mathematical models can help in predicting the evolution of a complex system such as the molecular circadian clock and are useful to discriminate between probable key nodes and accessory pathways. More generally, how mathematical models can provide the missing bridge between in vivo and in vitro settings has generally been underappreciated. Let us stress again that the same molecular network underlies the circadian clock in both cases, although the presence of important actors may differ between them. Therefore, it is conceivable that the mathematical model is trained and refined using an abundance of data from cell experiments, and then later adjusted to in vivo data for physiological relevance. This can greatly reduce the burden of going back and forth between modeling and experiments, and accelerate the development of predictive models.

Moreover, the time has come not only to decipher the clock but also to act on it. We have seen that several compounds are promising to reset a disturbed clock and reduce symptoms in several pathologies. Still, proper timing for the administration of such drugs will be critical to optimizing their beneficial effects. When we applied REV-ERB agonist pulses in our mathematical model for dampened clock oscillations, the timing of administration highly modified our ability to restore normal oscillation profiles or adverse effects [105]. In the same vein, phenolic compounds such as resveratrol have also been investigated for long as dietary supplements for their antioxidant properties, which were shown to depend on the time of administration [110].

In the field of oncology, pioneering works from Levi et al. demonstrated the benefit of chronotherapy in the treatment of cancers, by showing that a chronomodulated delivery of oxaliplatin, fluorouracil, and folinic acid improved the tolerance to treatments and the objective response of patients with metastatic colorectal cancers when compared to a constant infusion rate protocol [111].

More recently, researchers have evidenced circadian evolutions in the expression and activity levels of the P-glycoprotein, a key protein in the efflux of multiple drugs by cancer cells [112]. Besides, it was shown that trastuzumab-resistant gastric cancer cells displayed PPARγ- and PER1-controlled circadian fluctuations in their glycolytic activity [113]. Interestingly, PER1 silencing alleviated the resistance to trastuzumab. Altogether, these data strongly argue for the necessity to consider chronotherapeutics with much attention in the cancer field. In that frame, some pioneer studies have already demonstrated that the anti-tumor efficacy of compounds could be highly variable depending on the time of administration. For example, the pharmacological administration of metformin at ZT6 in the gastric cancer model aforementioned significantly improved trastuzumab efficacy [113]. In another setting, namely a breast cancer xenograft model, the efficacy of erastin, an inhibitor of xCT cystine/glutamate transporter, was improved when administered during the light phase when compared to the dark phase [114]. In the coming years, pharmacokinetics studies should take this into account, to define the optimal timing of anticancer drug delivery [115].

More globally, chronopharmacology is gaining importance for several pathologies, and trials are now evaluating the optimal timing for drug administration [116,117,118]. In the field of NASH (Non-Alcoholic SteatoHepatitis) particularly, many pharmacological targets are known to be involved both in metabolism and in the clock, calling for a need to consider chronopharmacology in the design of clinical trials, which may limit the variability of trial outcomes that have been observed so far [119]. We are convinced that mathematical models will be valuable to predict the time relevance of pharmacological intervention for optimal chronotherapy, to facilitate the transition from bench to bedside. In this context, the development of machine learning should prove very useful [120].

Once the optimal schedules for drug delivery are defined, it will be instrumental to have tools adequate to ensure proper timing of delivery. In this context, several strategies were explored to take the benefit of nanotechnologies for chronopharmacotherapy [121]. Porous beads have thus been investigated for their potential to offer a pulsatile drug delivery [122], still to be adapted for long periods. About circadian interventions, the development of delivery systems that can be implanted and remotely controlled [123] should be of great interest for daily administrations at specific times.

Finally, since patients may suffer from distinct clock misalignments, it will also be important to define with clear biomarkers their circadian state [124]. Several tools are now available to monitor the internal circadian time of patients thanks to gene expression analysis of blood samples [125,126], or epidermal samples [127], which will be instrumental for personalized medicine.

## 10. Conclusions

To sum up, an increasing number of studies strongly suggest that taking into account human circadian rhythms has tremendous potential to improve the human quality of a healthy life. This can be achieved by both developing daily good habits, such as avoiding food at night and redesigning shifting work and activity schedules and shifting some medical practices. In clinics, adjusting the time of surgery interventions may alleviate their morbidity. Moreover, the timing of drug administration against pathologies should as well benefit from a similar rethinking.

This will require much work to identify the optimal schedule among the many possible, but an interdisciplinary approach may help to identify the best time windows, and we are convinced that improving human health is worth the effort. Compelling data and tools are available; it is now time to integrate them thanks to a combination of efforts from all communities.

## Figures and Tables

**Figure 1 pharmaceutics-14-01424-f001:**
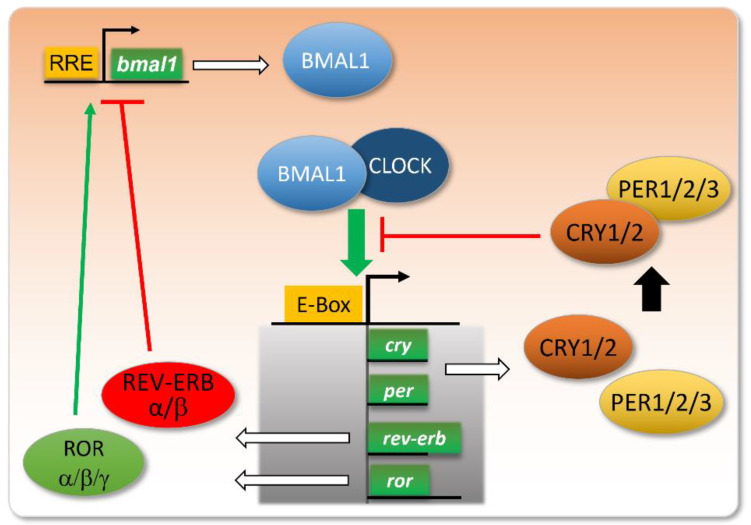
Core clock genes and their intertwined feedback loops.

**Figure 2 pharmaceutics-14-01424-f002:**
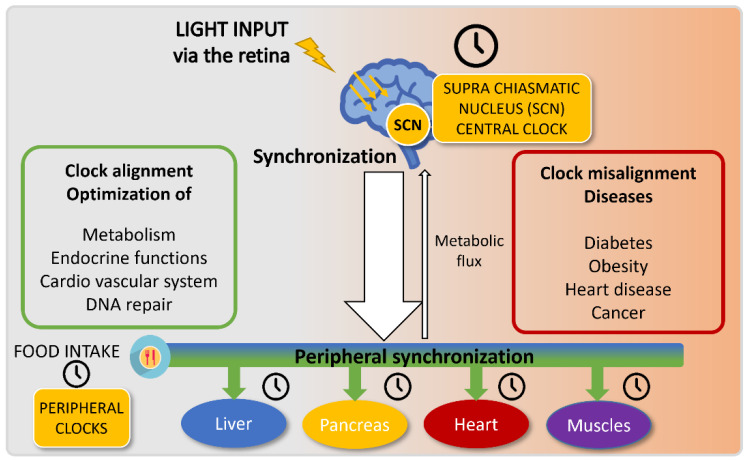
Circadian clock entrainment.

**Figure 3 pharmaceutics-14-01424-f003:**
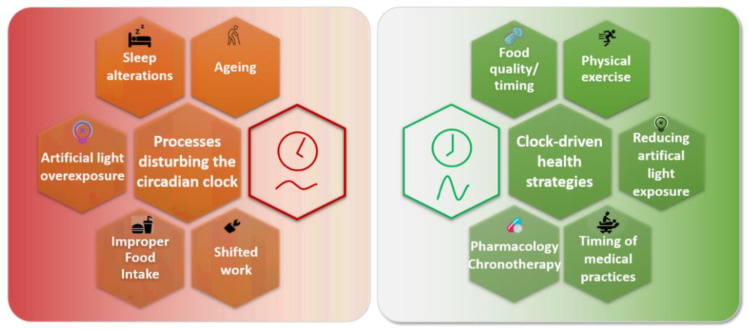
Restoring a proper alignment between the human circadian clock, metabolism, and activities.

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
