# Peer review of "An Overview of the Circadian Clock in the Frame of Chronotherapy: From Bench to Bedside"

_pharmaceutics, 2022, doi:10.3390/pharmaceutics14071424_

Round 1

Reviewer 1 Report

In this paper, the authors are discussing the links between circadian clocks and pathologies. The authors also demonstrate how chronotherapy should be integrated in health care, medicine and pharmacotherapy. In addition, the authors discuss the current challenges that the circadian biology faces and the tools to address them. The paper is written nicely, however, a few points should be addressed:

1.       There are some grammatical and phrasing issues that need to be revised throughout the text.

2.       Figure 1- why only Cry1 and Per2 are mentioned, while other isoforms of Per and Cry are also known to inhibit Clock and BMAL1 activity. Please update the figure accordingly.

3.       The sentence: “In the Gremlins movie, Gizmo’s 178 seller tells the hero not to give him food after midnight. Human beings should pay more 179 attention to this recommendation for themselves too” is very cute, but can also be misleading, since people reading it might think that only eating after midnight is not recommended, while the truth is that eating during resting phase is unhealthy, and the exact times my differ between people and are still poorly studied.

4.       A big portion that is missing in the text is circadian and cardiovascular disease. It was very briefly covered in one paragraph, however, since extensive research has been done on myocardial infarction and circadian recently, better coverage of this area of research should be included in this manuscript.

5.       Using “metabolism” as a key word and mentioning that this is the focus of the paper (in the abstract) is misleading, as metabolism is not the main area covered in this paper, but rather there is a wide coverage of pathologies. Please choose additional key words that would better reflect the text and change the abstract accordingly.

Author Response

Dear editor,

We thank the reviewers for their careful reading and suggestions, which have been taken into account to improve the manuscript.

A point-by-point response to their evaluation is given below.

We thank you very much for your consideration and look forward to hearing from you.

Yours sincerely,

Alessandro FURLAN                 Marc LEFRANC

Peer Reviewer #1

In this paper, the authors are discussing the links between circadian clocks and pathologies. The authors also demonstrate how chronotherapy should be integrated in health care, medicine and pharmacotherapy. In addition, the authors discuss the current challenges that the circadian biology faces and the tools to address them. The paper is written nicely, however, a few points should be addressed:

We thank the reviewer for her/his assessment and judgement of our work.

  1. There are some grammatical and phrasing issues that need to be revised throughout the text.

The manuscript has been revised and edited by an English language editing service.

  1. Figure 1- why only Cry1 and Per2 are mentioned, while other isoforms of Per and Cry are also known to inhibit Clock and BMAL1 activity. Please update the figure accordingly.

The figure has been modified according to the reviewer’s comment, and now includes the other Per/Cry isoforms that can impact CLOCK/BMAL1 activity.

  1. The sentence: “In the Gremlins movie, Gizmo’s 178 seller tells the hero not to give him food after midnight. Human beings should pay more 179 attention to this recommendation for themselves too” is very cute, but can also be misleading, since people reading it might think that only eating after midnight is not recommended, while the truth is that eating during resting phase is unhealthy, and the exact times my differ between people and are still poorly studied.

The sentence has been rephrased to take into account the potential confusion and misinterpretation, and now states: “Human beings should also pay more attention to the recommendation not to eat during their resting phase”.

  1. A big portion that is missing in the text is circadian and cardiovascular disease. It was very briefly covered in one paragraph, however, since extensive research has been done on myocardial infarction and circadian recently, better coverage of this area of research should be included in this manuscript.

We have now integrated more data on cardiovascular diseases, in sections 2 and 5 of our manuscript.

  1. Using “metabolism” as a key word and mentioning that this is the focus of the paper (in the abstract) is misleading, as metabolism is not the main area covered in this paper, but rather there is a wide coverage of pathologies. Please choose additional key words that would better reflect the text and change the abstract accordingly.

We agree with the reviewer and removed in the abstract the explicit reference to metabolism as a point of focus. We also added other keywords to better illustrate the topics of interest of this review.

Reviewer 2 Report

Review is dedicated to a topical problem of chronotherapy, is written in a plain language and can be reader-friendly for broader readership and newcomers to the topic. The following issues can be considered by authors:

1. Page 3. Figure 2. Though feeding indeed is synchronizer capable of uncoupling rhythms, e.g., DOI: 10.1101/gad.183500, generalization that circadian rhythms in peripheral organs are entrained mainly by the food, L.91 (“The circadian central and peripheral clocks are entrained by two main cues: light and food intake, respectively”) is over-simplification. Light as a principal zeitgeber modulates activity and feeding; studies depicting meal timing to fulfil zeitgeber criteria as was put forward by Jürgen Aschoff in the 1950s are lacking (DOI: 10.1038/s41574-020-0318-z). Recent studies showed that light and melatonin may impact metabolism independently of feeding (DOI: 10.1038/s41467-022-28308-6); DOI: 10.3390/cells9020489).

2. P.3 While discussing widespread modern factors that exert misaligning effects such as light and food, authors may wish to add that coffee as well was shown to have time-dependent effects, similar to light, turning from beneficial in the morning to disruptive after noon (DOI: 10.1126/scitranslmed.aac5125; DOI: 10.5664/jcsm.3170).

3. P.4., L.126 Sentence “And people who strongly differ from the norm suffer from a so-called “social jetlag” should be rephrased or explained: even though there are numerous recent publications linking health risks with social-jet-lag-prone, slow clocks, or later chronotypes, labelling such individuals as “differ from the norm” is not justified, unless they are defined as representatives of delayed sleep phase syndrome. Authors may consider adding discussion on papers showing associations between evening chronotypes with morbidity and mortality (e.g., DOI: 10.1080/07420528.2018.1454458; DOI: 10.1080/07420528.2020.1732403). Furthermore, please note the association between social jet lag and depressive symptoms (https://doi.org/10.1093/sleep/zsz204). On the other hand, associations between chronotype and somatic vs. chronotype vs. psychiatric disorders may differ (DOI: 10.1080/07420528.2018.1454458), being less obvious for the latter (DOI: 10.1016/j.bcp.2021.114438).

4. The authors are prompted to pay attention to fundamental differences in chronotype (habitual sleep preference, or midsleep) that advance with age, and intrinsic clock speed, endogenous tau, that basically remains unchanged (DOI: 10.1073/pnas.1010666108; DOI: 10.1016/j.jsmc.2015.08.002). Also, the term “chronotype” should not be confused with the phase of any objective circadian phase marker (e.g., melatonin, temperature, cortisol, or actimetry-derived), since their measuring tools are distinct and observed estimates are usually different.

5. In this context, term “chronotype” should be used with caution while discussing age-related changes (P4., L.135): “older people tend to have an early chronotype” in order not to mix sleep-wake phase with intrinsic clock speed. Authors also may wish to note that in age-related pathologies in which daylight perception is compromised (e.g., neurodegenerative processes such as glaucoma (DOI: 10.1080/07420528.2019.1566741), or Alzheimer’s diseases (DOI: 10.1176/appi.ajgp.13.5.359) circadian rhythms, but not chronotype(!), can be phase delayed, and thus phases of different rhythms can be delayed unequally (DOI: 10.3390/ijms22010359). Also, some gene SNPs may distinguish whether the changes of circadian parameters would occur or not under the presence of misaligning factor (DOI: 10.1111/jpi.12730). Authors may wish to add these discussions and also effects of melatonin in the elderly also on P.7., (Clock and age-related diseases).

6. P.5. While discussing healthy ways of life via synchronization with circadian clocks, authors may wish to introduce briefly the concept of human-centric-lighting (e.g., DOI: 10.3389/fneur.2021.630553; DOI: 10.3389/fpsyt.2021.652161).

7. P.8. “Current challenges and perspectives: how to define the most appropriate timing for 352 phamacotherapy”. Authors may wish to add more information regarding pioneering works in chronotherapy. For example, works of Dr. Francis Levi et al., for example most notable were discussed in an already cited work DOI: 10.1016/j.cmet.2019.06.019).

8. P.10., L.452. Please note that besides tools of circadian phase assessment using a single blood sample, there are also tools based on even less invasive epidermis samples (DOI: 10.1073/pnas.1809442115; DOI: 10.1186/s13073-020-00768-9).

Author Response

Peer Reviewer #2

Review is dedicated to a topical problem of chronotherapy, is written in a plain language and can be reader-friendly for broader readership and newcomers to the topic. The following issues can be considered by authors:

We thank the reviewer for her/his positive assessment and suggestions, very helpful to reinforce our messages.

  1. Page 3. Figure 2. Though feeding indeed is synchronizer capable of uncoupling rhythms, e.g., DOI: 10.1101/gad.183500, generalization that circadian rhythms in peripheral organs are entrained mainly by the food, L.91 (“The circadian central and peripheral clocks are entrained by two main cues: light and food intake, respectively”) is over-simplification. Light as a principal zeitgeber modulates activity and feeding; studies depicting meal timing to fulfil zeitgeber criteria as was put forward by Jürgen Aschoff in the 1950s are lacking (DOI: 10.1038/s41574-020-0318-z). Recent studies showed that light and melatonin may impact metabolism independently of feeding (DOI: 10.1038/s41467-022-28308-6); DOI: 10.3390/cells9020489).

We agree with the reviewer and modified the text and the figure legend accordingly to these comments.

  1. P.3 While discussing widespread modern factors that exert misaligning effects such as light and food, authors may wish to add that coffee as well was shown to have time-dependent effects, similar to light, turning from beneficial in the morning to disruptive after noon (DOI: 10.1126/scitranslmed.aac5125; DOI: 10.5664/jcsm.3170).

We thank the reviewer for this suggestion that we integrated into the impact of modern life on human circadian clock misalignment.

  1. P.4., L.126 Sentence “And people who strongly differ from the norm suffer from a so-called “social jetlag” should be rephrased or explained: even though there are numerous recent publications linking health risks with social-jet-lag-prone, slow clocks, or later chronotypes, labelling such individuals as “differ from the norm” is not justified, unless they are defined as representatives of delayed sleep phase syndrome. Authors may consider adding discussion on papers showing associations between evening chronotypes with morbidity and mortality (e.g., DOI: 10.1080/07420528.2018.1454458; DOI: 10.1080/07420528.2020.1732403). Furthermore, please note the association between social jet lag and depressive symptoms (https://doi.org/10.1093/sleep/zsz204). On the other hand, associations between chronotype and somatic vs. chronotype vs. psychiatric disorders may differ (DOI: 10.1080/07420528.2018.1454458), being less obvious for the latter (DOI: 10.1016/j.bcp.2021.114438).

We adjusted the text to emphasize these interesting correlations between chronotypes, social jetlag and psychological disorders.

  1. The authors are prompted to pay attention to fundamental differences in chronotype (habitual sleep preference, or midsleep) that advance with age, and intrinsic clock speed, endogenous tau, that basically remains unchanged (DOI: 10.1073/pnas.1010666108; DOI: 10.1016/j.jsmc.2015.08.002). Also, the term “chronotype” should not be confused with the phase of any objective circadian phase marker (e.g., melatonin, temperature, cortisol, or actimetry-derived), since their measuring tools are distinct and observed estimates are usually different.

We agree with the reviewer, but we found it difficult to develop these notions in any part of our review, without making an important digression. Such aspects are partially covered by the review to which we referred (Hood et Amir, 2017), and we added the reference suggested by the reviewer (Duffy et al, 2015).

  1. In this context, term “chronotype” should be used with caution while discussing age-related changes (P4., L.135): “older people tend to have an early chronotype” in order not to mix sleep-wake phase with intrinsic clock speed. Authors also may wish to note that in age-related pathologies in which daylight perception is compromised (e.g., neurodegenerative processes such as glaucoma (DOI: 10.1080/07420528.2019.1566741), or Alzheimer’s diseases (DOI: 10.1176/appi.ajgp.13.5.359) circadian rhythms, but not chronotype(!), can be phase delayed, and thus phases of different rhythms can be delayed unequally (DOI: 10.3390/ijms22010359). Also, some gene SNPs may distinguish whether the changes of circadian parameters would occur or not under the presence of misaligning factor (DOI: 10.1111/jpi.12730).

Authors may wish to add these discussions and also effects of melatonin in the elderly also on P.7., (Clock and age-related diseases).

We thank the reviewer for this comment, which prompted us to integrate these notions of misalignment of phases between distinct features.

  1. P.5. While discussing healthy ways of life via synchronization with circadian clocks, authors may wish to introduce briefly the concept of human-centric-lighting (e.g., DOI: 10.3389/fneur.2021.630553; DOI: 10.3389/fpsyt.2021.652161).

We thank the reviewer for this suggestion, which we included when discussing light interventions.

  1. P.8. “Current challenges and perspectives: how to define the most appropriate timing for 352 phamacotherapy”. Authors may wish to add more information regarding pioneering works in chronotherapy. For example, works of Dr. Francis Levi et al., for example most notable were discussed in an already cited work DOI: 10.1016/j.cmet.2019.06.019).

We focused in the previous version of the manuscript on the recent evidence for a benefit of chronotherapy in oncology, to which Dr Francis Lévi highly contributes. We included in the new version some information about his pioneering works as well.

  1. P.10., L.452. Please note that besides tools of circadian phase assessment using a single blood sample, there are also tools based on even less invasive epidermis samples (DOI: 10.1073/pnas.1809442115; DOI: 10.1186/s13073-020-00768-9).

These actually constitute very promising tools, that we mention in this new version of the manuscript.

Round 2

Reviewer 2 Report

The authors provided adequate answers to the stated issues and improved the manuscript.

This manuscript is a resubmission of an earlier submission. The following is a list of the peer review reports and author responses from that submission.